# Pharmacokinetics of a 503B outsourcing facility-produced theophylline in dogs

**Jennifer M. Reinhart**[1]*, **Gabriela A. R. de Oliveira**[1], **Lauren Forsythe**[1], **Zhong Li**[2]

**1** Department of Veterinary Clinical Medicine, College of Veterinary Medicine, University of Illinois Urbana-Champaign, Urbana, IL, United States of America, **2** Metabolomics Lab, Roy J. Carver Biotechnology Center, University of Illinois Urbana-Champaign, Urbana, IL, United States of America

* jreinha2@illinois.edu

**Data Availability Statement:** All relevant data are within the paper and its Supporting Information files.

**Funding:** The authors declare that this study received funding from Epicur® Pharma

## Abstract

Theophylline is an important drug for treatment of canine chronic bronchitis and bradyarrhythmias, but new products require validation since pharmacokinetics in dogs can vary by formulation. A new, 503B outsourcing facility-produced theophylline product (OFT) is available for veterinary use. Outsourcing facilities have many advantages over traditional compounding sources including current good manufacturing practice compliance. The purpose of this study was to establish the pharmacokinetics of OFT in dogs. Eight healthy dogs received 11 mg/kg intravenous aminophylline and 10 mg/kg oral OFT followed by serial blood sampling in a two-way, randomized, crossover design with 7-day washout. Plasma theophylline concentrations were quantified by liquid chromatography-mass spectrometry. Bioavailability, maximum concentration, time to maximum concentration, half-life and area under the curve were: 97 ± 10%, 7.13 ± 0.71 μg/mL, 10.50 ± 2.07 h, 9.20 ± 2.87 h, and 141 ± 37.6 μg*h/mL, respectively. Steady-state predictions supported twice daily dosing of the OFT, but specific dosage recommendations are hindered by lack of a canine-specific therapeutic range for plasma theophylline concentration. These findings suggest that the OFT is well absorbed and can likely be dosed twice daily in dogs, but future pharmacodynamic and clinical studies are needed to establish a definitive therapeutic range for theophylline in this species.

## Introduction

Theophylline is a methylxanthine drug used as a bronchodilator in the treatment of canine chronic bronchitis and as a therapy for certain bradyarrhythmias [1, 2]. Although pharmacokinetics have been established in dogs for several oral theophylline formulations in the past [3–7], most of these are no longer commercially available, likely due to the decline of theophylline use in human medicine [1]. Other products approved for use in humans are available, but it is important to validate individual products in dogs because pharmacokinetic parameters, particularly bioavailability, differ between formulations [6].

One alternative to human-approved theophylline products, which seem to go on and off the market frequently, is veterinary compounding. Traditional compounding requires drugs

(epicurpharma.com). The funder was allowed review of the manuscript prior to submission but was not involved in the study design, collection, analysis, interpretation of data, the writing of this article or the decision to submit it for publication.

**Competing interests:** The authors have declared that no competing interests exist.

to be made in small batches for individual patients, following United States Pharmacopeia (USP) standards, which may raise concerns regarding product consistency, safety and efficacy due to the lack of testing requirements in many instances. In 2013, the Food and Drug Administration (FDA) created a new category of facilities: the 503B outsourcing facility [8], which requires these facilities to follow current good manufacturing practice (cGMP) regulations resulting in a safe and consistent product. Furthermore, these facilities must register with and receive oversight from the FDA. Thus, 503B outsourcing facilities may represent a new, high-quality source of theophylline for veterinary use. However, pharmacokinetics of such products must be established prior to use in order to guide dosing recommendations. The purpose of this study was to establish oral, single-dose pharmacokinetic parameters of a 503B outsourcing facility-produced theophylline (OFT) in dogs.

## Materials and methods

Eight healthy dogs were recruited from the pet population of the students, faculty and staff of the University of Illinois, College of Veterinary Medicine (S1 Table). Written, informed consent was obtained from owners of all animals included. Animals were considered healthy based on a physical exam performed by a board-certified internist (JR), complete blood count, serum biochemistry panel, urinalysis, and total serum thyroxine concentration. At the time of the study and for at least two weeks prior, dogs were not taking any medications other than routine flea, tick, and heartworm prophylaxis. This study was approved by the University of Illinois Institutional Animal Care and Use Committee (Protocol #20030).

The OFT was obtained from a 503B outsourcing facility in 25, 50, 75, and 100 mg tablets (Theophylline ER Mini- and Mighty-Med Triangles; Epicur Pharma®, Mount Laurel, NJ). This facility maintains cGMP requirements by observing Code of Federal Regulations (CFR) Title 21, Part 210 [9] and Part 211 [10]. The OFT is formulated with an excipient having a polymeric backbone of cellulose in a specific ratio designed to regulate the release of theophylline from the tablet. Preliminary testing has shown that the product used in this study was of appropriate concentration and was expected to be stable throughout the study period. Full stability of the product testing in accordance with CFR 21, Part 211 is currently underway.

This study was a two-way crossover design with an intravenous phase and an oral phase, separated by a 7-day washout period. This washout period was selected based on a ~12 h half-life reported for other theophylline products in dogs [4, 7] leading to > 99.9% elimination of a single dose prior to the next phase. Dogs were randomized to undergo either the intravenous phase (n = 4) or oral phase (n = 4) first. At least 15 hours prior to each phase, at least 15 hours prior, central venous catheters were placed in the jugular vein by the modified Seldinger technique [11] under sedation (2–4 ug/kg dexmedetomidine IV [Zoetis Inc., Parsippany-Troy Hills, NJ], 0.2 mg/kg butorphanol IV [Zoetis Inc.], 0.02–0.04 mg/kg atipamezole IM reversal [Zoetis Inc.]). For the intravenous phase, a peripheral catheter was also placed in a cephalic vein for IV aminophylline administration. Dogs had free access to water throughout the study and were fed their normal diet in twice daily rations. On the first morning of each phase, the diet was fed just after drug administration. In the intravenous phase, dogs were administered a single 11 mg/kg IV dose of aminophylline, the ethylenediamine salt of theophylline (8.6 mg/kg theophylline equivalent, Hospira, Lake Forest, IL), over 15–30 s followed by a saline flush. In the oral phase, dogs were administered a single ~10 mg/kg oral dose of the OFT with a small amount of wet food, rounded to the nearest 25 mg-increment to account for tablet sizes.

For the intravenous phase, blood samples were collected at 0, 2, 5, 15, 30, and 45 min, and 1, 2, 4, 8, 12 and 24 h. For the oral phase, blood samples were collected at 0, 15, 30, and 45 min, and 1, 1.5, 2, 4, 8, 12, 18, 24, and 48 h. All samples were collected via the jugular catheter except

the 48 h oral phase sample, which was collected via peripheral venipuncture. Immediately after collection, blood was transferred to dipotassium EDTA tubes and stored at 4˚C. Within 4 h of collection, samples were centrifuged at 1,800 $x$ $g$ at 4˚C and plasma stored at -80˚C until analysis. Plasma theophylline concentrations were measured using a previously validated liquid chromatography-tandem mass spectrometry assay with lower limit of quantification of 2 ng/mL and intra- and inter- assay coefficients of variation of 1.3–3.8% and 1.6–4.1%, respectively, in the assay range relevant to this study [7, 12].

Continuous data are presented as mean ± standard deviation. Non-compartmental pharmacokinetic analysis was performed using Phoenix WinNonlin (Certara L.P., Princeton, NJ) for both the intravenous and oral phases. Absolute bioavailability of the OFT was calculated for each animal by: $F = (AUC_{PO} * Dose_{IV}) / (AUC_{IV} * Dose_{PO})$ x 100%, where AUC is the area under the curve extrapolated to infinity. Mean absorption time (MAT) was calculated for each animal as the difference between the mean residence times (MRTs) for the oral and intravenous phases. Steady state predictions of theophylline concentrations following twice daily, 10 mg/kg OFT administration were made using the NonParametric Superposition function in WinNonlin, which assumes linear pharmacokinetics are present for the investigated drug. Predicted plasma concentrations were compared to the therapeutic range established for adult humans (10–20 μg/mL) [13] as well as a proposed range for dogs of 5–30 μg/mL [12]. To evaluate for potential "flip-flop" kinetics and extended-release properties of the OFT, the terminal rate constant ($\lambda_z$) was compared between the intravenous and oral phases using a paired t-test in Prism 8 (GraphPad Software Inc., San Diego, CA). Significance was set at p < 0.05.

## Results

All 8 dogs recruited successfully completed the study. Summary statistics for pharmacokinetic parameters for intravenous aminophylline and oral OFT are presented in Table 1 and concentration-time curves are presented in Fig 1. Plasma theophylline concentrations for individual dogs for the intravenous and oral phases are presented in S2 and S3 Tables, respectively. Pharmacokinetic parameters for individual dogs for the intravenous and oral phases are presented in S4 and S5 Tables, respectively.

Fig 2 presents steady state predictions of plasma theophylline concentrations following 10 mg/kg OFT administered orally twice daily to dogs. S1 Fig. presents prediction data for individual participants. Steady state plasma theophylline concentrations were predicted to remain within the 10–20 μg/mL range for 56.9 ± 43.5% of the dosing interval with 3/8 dogs remaining within the range for 100% of the dosing interval and 1/8 dogs remaining below the range for 100% of the dosing interval. Steady state plasma theophylline concentrations were predicted to remain within the 5–30 μg/mL range for 99.3 ± 2.1% of the dosing interval with 7/8 dogs remaining within the range for 100% of the dosing interval. No dogs were predicted to have plasma concentrations exceeding 20 μg/mL at any time using 10 mg/kg OFT.

There was no significant difference in $\lambda_z$ between intravenous aminophylline and oral OFT administration (p = 0.472, Fig 3).

## Discussion

Despite its use in the treatment of canine chronic bronchitis and certain canine bradyarrhythmias, there are no theophylline formulations FDA-approved for use in dogs. Several previous studies have validated the use of human theophylline products in dogs [3–6], but most of these are no longer commercially available in the United States. In fact, the only product that has been evaluated in dogs and can still be purchased, Theo-24 (Endo Pharmaceuticals Inc.,

**Table 1. Pharmacokinetic parameter summary statistics for non-compartmental analysis of single dose intravenous aminophylline and oral OFT in dogs.**

| Parameter | Intravenous Aminophylline | | | | Oral OFT | | | |
|---|---|---|---|---|---|---|---|---|
| | Mean | SD | Min | Max | Mean | SD | Min | Max |
| D (mg/kg) | 8.74 | 0.45 | 8.41 | 9.77 | 10.26 | 0.55 | 9.74 | 11.49 |
| $\lambda_z$ (h$^{-1}$) | 0.085 | 0.021 | 0.045 | 0.106 | 0.081 | 0.024 | 0.047 | 0.127 |
| $t_{1/2}$ (h) | 8.75 | 2.97 | 6.57 | 15.27 | 9.20 | 2.87 | 5.47 | 14.84 |
| $C_0$ (µg/mL) | 19.00 | 2.87 | 16.38 | 23.74 | - | - | - | - |
| $T_{MAX}$ (h) | - | - | - | - | 10.50 | 2.07 | 8.00 | 12.00 |
| $C_{MAX}$ (µg/mL) | - | - | - | - | 7.13 | 0.71 | 6.23 | 8.01 |
| $AUC_{obs}$ (µg*h/mL) | 105.1 | 18.9 | 77.7 | 136.2 | 132.6 | 32.1 | 74.3 | 184.6 |
| $AUC_{0-\infty}$ (µg*h/mL) | 123.7 | 31.1 | 83.1 | 179.7 | 141.1 | 37.6 | 81.7 | 209.7 |
| $AUC_{0-\infty}$/D (µg*h/mL)/(mg/kg) | 14.2 | 3.9 | 9.8 | 21.4 | 13.8 | 3.6 | 8.4 | 20.4 |
| $AUC_{\%Extrap}$ (%) | 13.5 | 8.3 | 6.5 | 31.6 | 5.7 | 3.7 | 2.2 | 11.9 |
| $V_z$ (mL/kg) | 892.8 | 158.2 | 617.9 | 1059.7 | - | - | - | - |
| Cl (mL/kg/h) | 74.7 | 19.2 | 46.8 | 101.6 | - | - | - | - |
| $AUMC_{obs}$ (µg*h$^2$/mL) | 785.0 | 211.0 | 468.6 | 1100.4 | 2256.1 | 776.3 | 793.3 | 3262.9 |
| $AUMC_{0-\infty}$ (µg*h$^2$/mL) | 1525.9 | 962.1 | 648.7 | 3716.5 | 2778.0 | 1211.3 | 1027.6 | 5001.0 |
| $AUMC_{\%Extrap}$ (%) | 41.1 | 14.5 | 27.8 | 70.4 | 16.7 | 9.7 | 7.6 | 34.8 |
| MRT (h) | 11.59 | 4.08 | 7.80 | 20.68 | 18.88 | 3.69 | 12.58 | 23.85 |
| MAT (h) | - | - | - | - | 7.29 | 2.62 | 3.17 | 9.75 |
| F (%) | - | - | - | - | 97 | 10 | 85 | 114 |

The dosage listed for the intravenous aminophylline phase is presented as theophylline equivalent.

D = dosage; $\lambda_z$ = terminal rate constant; $t_{1/2}$ = terminal half-life; $C_0$ = calculated concentration at time 0 for intravenous phase; $T_{MAX}$ = time at maximum concentration; $C_{MAX}$ = maximum concentration; $AUC_{obs}$ = observed area under the curve; $AUC_{0-\infty}$ = AUC extrapolated to infinity; $AUC_{0-\infty}$/D = $AUC_{0-\infty}$ normalized to dosage; $AUC_{\%Extrap}$ = percent AUC extrapolated; $V_z$ = apparent volume of distribution during terminal phase; Cl = clearance; $AUMC_{obs}$ = observed area under the moment curve; $AUMC_{0-\infty}$ = AUMC extrapolated to infinity; $AUMC_{\%Extrap}$ = percent AUMC extrapolated; MRT = mean residence time; MAT = mean absorption time; F = bioavailability.

Malvern, PA), has poor bioavailability and is not recommended in this species [6]. Therefore, veterinarians must look to alternative sources for validated and consistently available products.

In this study, we established the single-dose pharmacokinetics of the oral theophylline product (OFT) manufactured by an FDA-registered, veterinary focused 503B outsourcing facility. The OFT demonstrated high bioavailability (97 ± 10%) with relatively low inter-individual variability (range 85–114%) in our study population. It also had a fairly long terminal half-life (9.20 ± 2.87 h), suggesting twice daily dosing may be appropriate to maintain plasma theophylline concentrations. This half-life is similar to what has been reported for other oral theophylline products in non-Beagle dogs (8.7–12.7 h). Early reports of short half-lives for rapid-release theophylline formulations (5.7 h) led to the original recommendation for using human extended-release products in dogs to allow twice daily dosing [5, 6, 14]. However, it is now suspected that these shorter half-lives were the result of increased intrinsic clearance of theophylline by the purpose-bred animals used in these studies rather than an effect of drug formulation [4]. This is evidenced by the fact that the terminal half-life of intravenous aminophylline in non-Beagle dogs is also fairly long (7.5–9.2 h), although a direct breed comparison has not been conducted [4, 7, 15]. The similarity of intravenous and oral terminal half-lives for theophylline in previous studies suggest that the products investigated do not possess extended-release properties in dogs [4, 7]. The same appears to be true for the OFT as we did not identify a significant difference in terminal half-lives between phases in our study

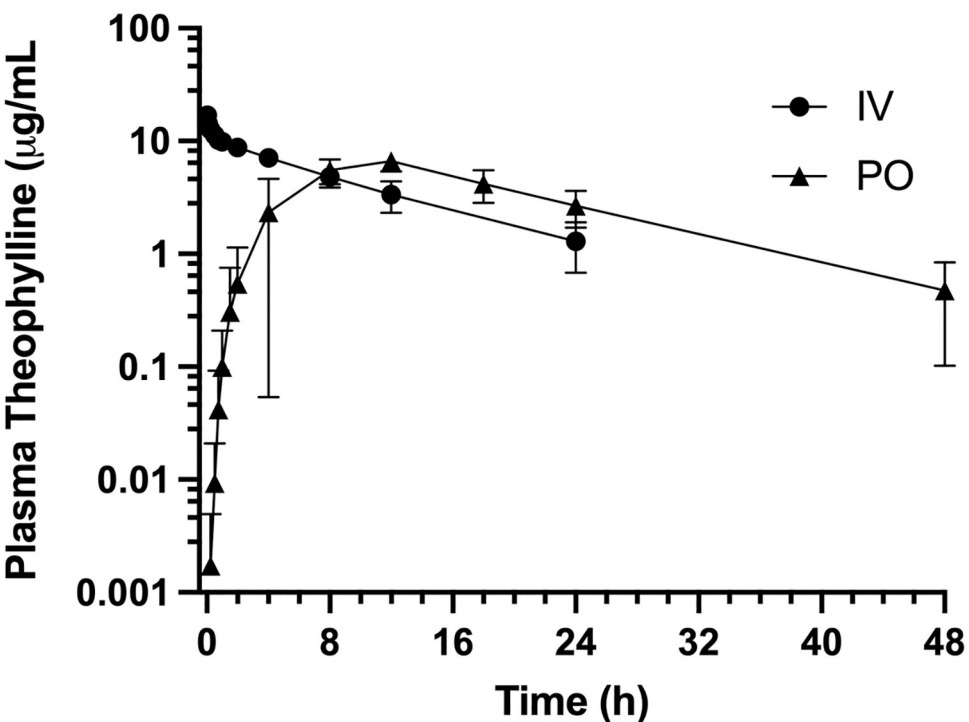

**Fig 1. Concentration-time curves.** Concentration-time curves of intravenously administered aminophylline (11 mg/kg) and orally administered OFT. Circles represent IV aminophylline and triangles represent PO OFT.

(8.75 ± 2.97 h vs. 9.20 ± 2.87 h, p = 0.472). Regardless, twice daily administration appears to be appropriate based on its longer half-life and because of its delayed time to peak concentration ($T_{MAX}$ = 10.50 ± 2.07 h).

Several pharmacokinetic parameters in our study demonstrated moderate interindividual variation including $t_{1/2}$ in both the intravenous and oral phases. Half-life is a hybrid pharmacokinetic parameter, affected by both distribution and elimination. Thus, variation in $t_{1/2}$ can be explained by variation in $V_z$, Cl, or both. When examining variability using a coefficient of variation (%CV = SD/mean * 100%), %CV for these parameters were 17.7% and 25.7%, respectively. Thus, although both may contribute to interindividual variation of $t_{1/2}$, Cl appears to do so to a greater degree. Similar trends of higher variation in clearance compared to distribution have been reported in other pharmacokinetic studies of theophylline in dogs as well [4, 7, 15]. Interindividual variation in Cl may be due to differences in hepatic metabolism of theophylline by the cytochrome P450 enzyme system [15]. MAT also demonstrated moderate interindividual variability in our study. MAT is difference between the MRTs for the extravascular and intravascular routes investigated and reflects the rate of drug absorption. However, variation in absorption rate did not appear to affect the overall extent of absorption as evidenced by the high bioavailability of the OFT in all study participants. This is in contrast to a study of a different compounded theophylline product for dogs in which both MAT and *F* were highly variable [7]. Another report compared the canine pharmacokinetics of four different extended-release theophylline products approved for human use (three of which are no longer available) and found that the degree of variability in MAT differed between products with %CV ranging from 29.5% to 116% [6]. Thus, interindividual variation in MAT may be a function of the specific theophylline formulation.

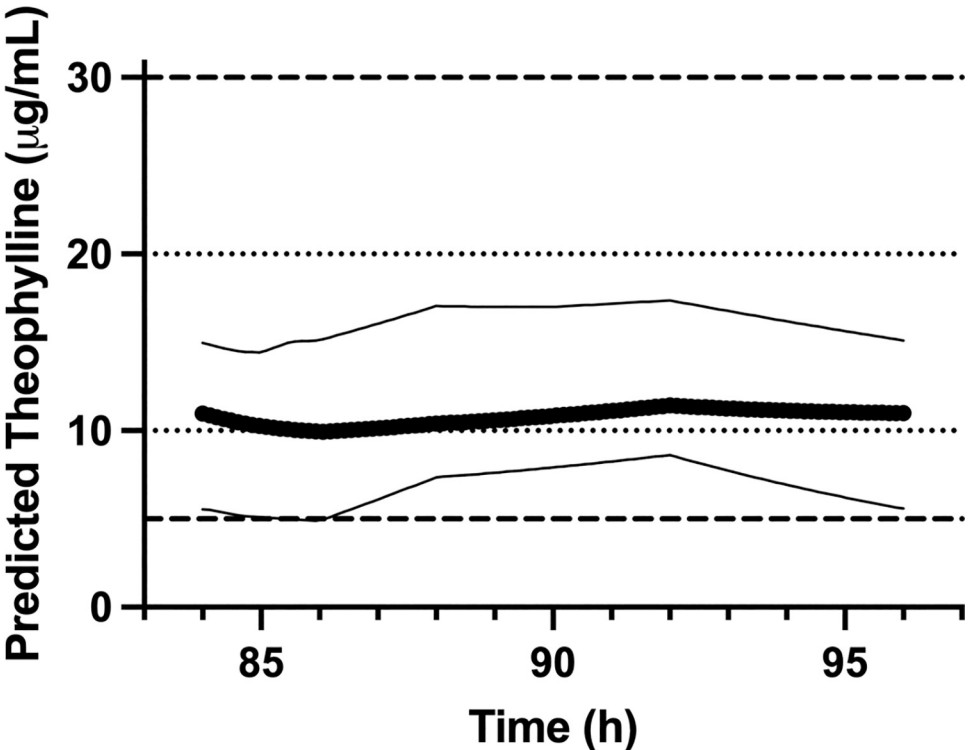

**Fig 2. Steady-state predictions.** Predicted steady-state plasma theophylline concentrations for twice daily administration of 10 mg/kg OFT. The thick solid line represents mean predicted concentrations for study participants (n = 8) and the thin solid lines represent the range. The dotted lines represent the theophylline therapeutic range established for adult humans (10–20 µg/mL) and the dashed lines represent a proposed range for use in dogs (5–30 µg/mL).

Our steady state predictions further support a 12 h dosing interval for the OFT. A recommendation for OFT dosage is more difficult to make because a validated therapeutic range has not been established for theophylline in dogs. Many previous studies have used 10–20 µg/mL, which is the target range recommended for adult humans [13]. To achieve consistent plasma concentrations above 10 µg/mL, a higher dosage would likely be needed since the dogs in our study were predicted to have concentrations below that level ~43% of the dosing interval on average, when dosed at 10 mg/kg. However, the 10–20 µg/mL theophylline therapeutic range used in humans may not be appropriate for dogs. We recently proposed a wider target range of 5–30 µg/mL for this species based on previous studies documenting increased ventilatory drive and tidal volumes above 5 µg/mL and lack of adverse effects below 37 µg/mL in dogs [12, 16, 17]. Targeting this range, 10 mg/kg OFT twice daily may be an appropriate dosing regimen since dogs in our study were predicted to be within the range for 99.3 ± 2.1% of the dosing interval using this dosage.

As for many initial, single-dose pharmacokinetics studies of drug products, one limitation of our study is its small sample size, which precludes thorough evaluation of population variability in OFT pharmacokinetic parameters. The use of client-owned dogs of various breeds and sizes may also be a limitation. These concerns may be best addressed using a population pharmacokinetics design and would be particularly interesting, given the suspected breed-based variability in theophylline elimination [4, 15]. In general, using compounded products in a study is also a potential limitation because drug potency, stability, and sterility cannot be guaranteed without testing individual batches. The use of a product from a 503B outsourcing

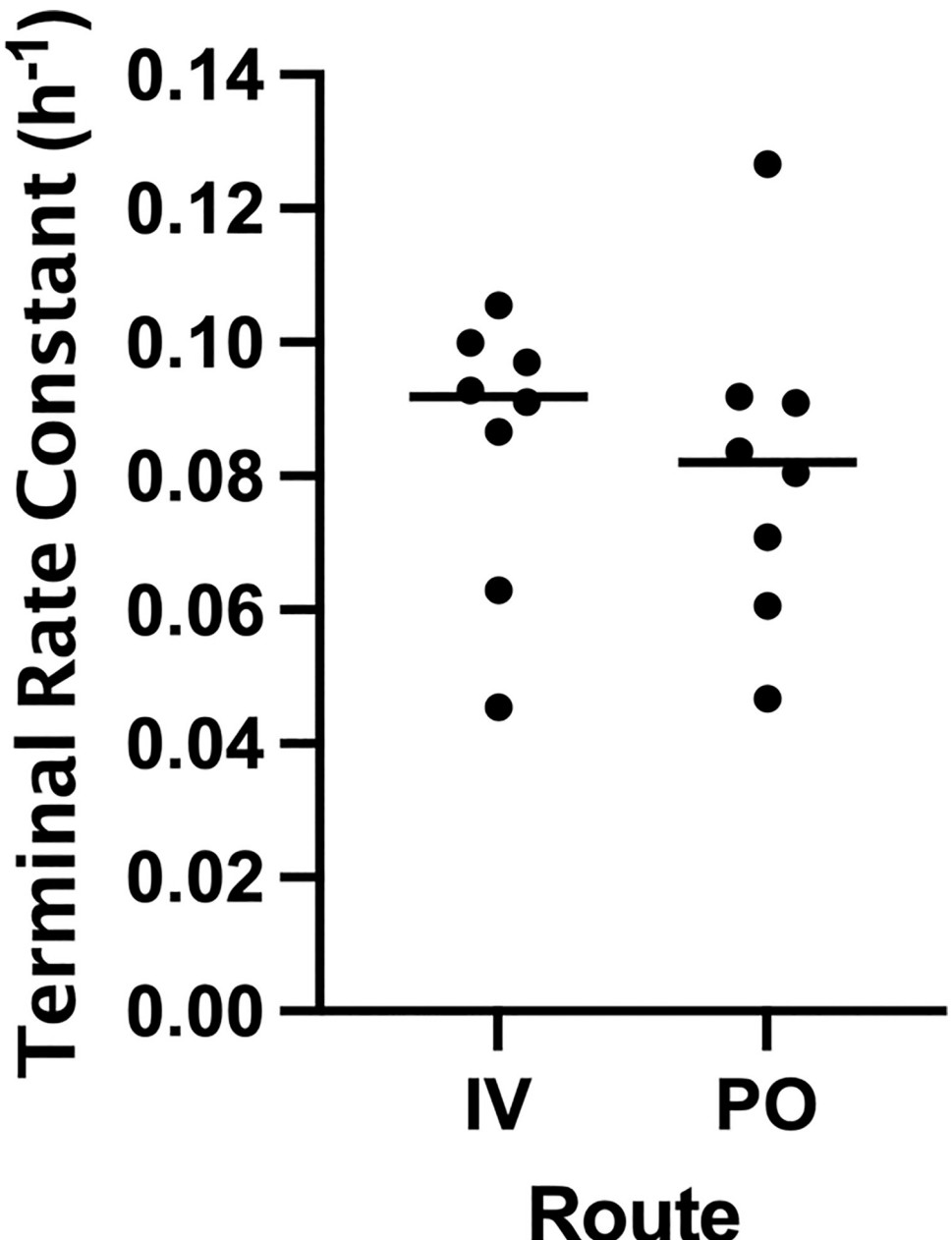

**Fig 3. Comparison of $\lambda_z$ between intravenous and oral phases.** There was no significant difference in $\lambda_z$ (p = 0.472) suggesting a lack of "flip-flop" kinetics or extended-release properties of the OFT.

facility largely addresses these concerns because these facilities adhere to cGMP and are routinely inspected by the FDA [8]. Another potential limitation is that the method used to predict steady state plasma theophylline concentrations following multidose OFT administration assumes linear pharmacokinetics, which has not been definitively demonstrated for theophylline in dogs. We recently completed a multidose pharmacokinetic study of a different compounded theophylline product and found that the half-life of theophylline does not significantly change at the higher plasma concentrations found after multiple doses [12]. This suggests that theophylline kinetics are linear and elimination processes are not saturated at

plasma concentrations investigated. However, theophylline plasma concentrations and kinetics should be investigated in a multidose study of the OFT product. Another limitation is the possibility for pharmacokinetic interactions between the sedatives administered and the aminophylline or OFT. In dogs, butorphanol, dexmedetomidine, and atipamezole have half-lives of 1.6, and < 3 hours, respectively [18–20]. These drugs were administered at least 15 hours before either test drug was given, so > 95% of the sedatives would have been eliminated by that time. However, a minor interaction cannot be completely ruled out. Finally, dosing recommendations for dogs prescribed any theophylline product are hindered by lack of a validated therapeutic range. Species-specific pharmacodynamic and clinical trials are needed.

## Conclusions

These data presented herein support that the OFT may be an appropriate source of theophylline for twice daily oral use in dogs. Dosages of 10 mg/kg are predicted to achieve concentrations that may be therapeutic, but future studies are needed. These findings are applicable to the 503B outsourcing facility-produced theophylline product investigated in this study. Theophylline products from other 503B outsourcing facilities would have a similar benefit of providing a consistent and reliable product but would individual validation in the species of interest.

## Supporting information

**S1 Table. Population demographics.** Demographic data for individual dogs participating in study. Dogs with study IDs beginning with 1 underwent the IV phase followed by the PO phase. Dogs with study IDs beginning with 2 underwent the PO phase followed by the IV phase. Weight represents the dog's weight at the study admission visit. FS = female spayed; MC = male castrated; MI = male intact.
(PDF)

**S2 Table. Plasma theophylline concentrations, intravenous phase.** Plasma theophylline concentrations for individual dogs following a single intravenous dose of 11 mg/kg aminophylline.
(PDF)

**S3 Table. Plasma theophylline concentrations, oral phase.** Plasma theophylline concentrations for individual dogs following a single oral dose of 10 mg/kg OFT.
(PDF)

**S4 Table. Individual pharmacokinetic parameters, intravenous phase.** Pharmacokinetic parameters from non-compartmental analysis of single dose intravenous aminophylline in individual dogs. The dosage listed is presented as theophylline equivalent. D = dosage; $\lambda_z$ = terminal rate constant; $t_{1/2}$ = terminal half-life; $C_0$ = calculated concentration at time 0 for intravenous phase; $AUC_{obs}$ = observed area under the curve; $AUC_{0-\infty}$ = AUC extrapolated to infinity; $AUC_{0-\infty}/D$ = $AUC_{0-\infty}$ normalized to dosage; $AUC_{\%Extrap}$ = percent AUC extrapolated; $V_z$ = apparent volume of distribution during terminal phase; Cl = clearance; $AUMC_{obs}$ = observed area under the moment curve; $AUMC_{0-\infty}$ = AUMC extrapolated to infinity; $AUMC_{\%Extrap}$ = percent AUMC extrapolated; MRT = mean residence time.
(PDF)

**S5 Table. Individual pharmacokinetic parameters, oral phase.** Pharmacokinetic parameters from non-compartmental analysis of single dose oral OFT in individual dogs. D = dosage; $\lambda_z$ = terminal rate constant; $t_{1/2}$ = terminal half-life; $T_{MAX}$ = time at maximum concentration;

$C_{MAX}$ = maximum concentration; $AUC_{obs}$ = observed area under the curve; $AUC_{0-\infty}$ = AUC extrapolated to infinity; $AUC_{0-\infty}/D$ = $AUC_{0-\infty}$ normalized to dosage; $AUC_{\%Extrap}$ = percent AUC extrapolated; $AUMC_{obs}$ = observed area under the moment curve; $AUMC_{0-\infty}$ = AUMC extrapolated to infinity; $AUMC_{\%Extrap}$ = percent AUMC extrapolated; MRT = mean residence time; MAT = mean absorption time; F = bioavailability.
(PDF)

**S1 Fig. Individual dog steady state predictions.** Steady statement plasma theophylline concentration predictions for individual dogs when administered 10 mg/kg OFT twice daily (thick solid line). The thin dotted lines represent the theophylline therapeutic range established for adult humans (10–20 μg/mL) and the thin dashed lines represent a proposed range for use in dogs (5–30 μg/mL).
(TIFF)

## Acknowledgments

The authors thank the owners of the dogs that participated in this study and thank Dr. Brendan McKiernan for assistance with manuscript preparation.

## Author Contributions

**Conceptualization:** Jennifer M. Reinhart, Lauren Forsythe, Zhong Li.

**Data curation:** Jennifer M. Reinhart.

**Formal analysis:** Jennifer M. Reinhart, Gabriela A. R. de Oliveira, Lauren Forsythe, Zhong Li.

**Funding acquisition:** Jennifer M. Reinhart, Lauren Forsythe.

**Investigation:** Jennifer M. Reinhart, Gabriela A. R. de Oliveira, Zhong Li.

**Methodology:** Jennifer M. Reinhart, Gabriela A. R. de Oliveira, Lauren Forsythe, Zhong Li.

**Project administration:** Jennifer M. Reinhart, Gabriela A. R. de Oliveira.

**Resources:** Jennifer M. Reinhart, Lauren Forsythe, Zhong Li.

**Software:** Jennifer M. Reinhart, Gabriela A. R. de Oliveira.

**Supervision:** Jennifer M. Reinhart.

**Validation:** Zhong Li.

**Visualization:** Jennifer M. Reinhart.

**Writing – original draft:** Jennifer M. Reinhart, Gabriela A. R. de Oliveira.

**Writing – review & editing:** Jennifer M. Reinhart, Gabriela A. R. de Oliveira, Lauren Forsythe, Zhong Li.

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
