## [Decision Letter · Decision Letter 0]

23 Nov 2021

PONE-D-21-32370Pharmacokinetics of a 503B outsourcing facility-produced theophylline in dogsPLOS ONE

Dear Dr. Reinhart,

Thank you for submitting your manuscript to PLOS ONE. After careful consideration, we feel that it has merit but does not fully meet PLOS ONE’s publication criteria as it currently stands. Therefore, we invite you to submit a revised version of the manuscript that addresses the points raised during the review process. Please submit your revised manuscript by Jan 07 2022 11:59PM. If you will need more time than this to complete your revisions, please reply to this message or contact the journal office at plosone@plos.org. Please include the following items when submitting your revised manuscript:A rebuttal letter that responds to each point raised by the academic editor and reviewer(s). You should upload this letter as a separate file labeled 'Response to Reviewers'.A marked-up copy of your manuscript that highlights changes made to the original version. You should upload this as a separate file labeled 'Revised Manuscript with Track Changes'.An unmarked version of your revised paper without tracked changes. You should upload this as a separate file labeled 'Manuscript'.

We look forward to receiving your revised manuscript.

Kind regards,

Kaisar Raza

Academic Editor

PLOS ONE

Journal Requirements:

2. In your Methods section, please provide additional details regarding participant consent from the owners of the animals. In the ethics statement in the Methods and online submission information, please ensure that you have specified (1) whether consent was informed and (2) what type you obtained (for instance, written or verbal). If the need for consent was waived by the ethics committee, please include this information.

Reviewers' comments:

Reviewer's Responses to Questions

**Comments to the Author**

1. Is the manuscript technically sound, and do the data support the conclusions?

Reviewer #1: Yes

Reviewer #2: Yes

Reviewer #3: Yes

2. Has the statistical analysis been performed appropriately and rigorously? 

Reviewer #1: Yes

Reviewer #2: Yes

Reviewer #3: Yes

3. Have the authors made all data underlying the findings in their manuscript fully available?

Reviewer #1: Yes

Reviewer #2: Yes

Reviewer #3: Yes

4. Is the manuscript presented in an intelligible fashion and written in standard English?

Reviewer #1: Yes

Reviewer #2: Yes

Reviewer #3: Yes

5. Review Comments to the Author

Reviewer #1: In this manuscript, the authors aimed to establish oral, single-dose pharmacokinetic parameters of a 503B outsourcing facility-produced theophylline (OFT) in dogs. The authors suggest that the OFT is well absorbed and can likely be dosed twice daily in dogs and the dose of 10 mg/kg are predicted to achieve concentrations that may be therapeutic.

The subject of the manuscript is original and falls within the scope of the journal. I think this study provides scientifically valuable data for the literature.

Some modifications are needed for the MS according to the points that I have indicated.

1.Lines 23-24: “… 503B outsourcing facility that manufactures veterinary drugs (OFT), …”, but in Lines 56-57: “… a 503B outsourcing facility-produced theophylline (OFT) in dogs”. Please revise the first abbreviation in the abstract.

2.Line 64: “At the time of the study…”. Before the study, the animals should not have taken any medication for a certain period!

3.Lines 75-79: “This study was conducted in two phases …” or two-way crossover?

4.Lines 79-82: There was a possibility of pharmacokinetic interaction between these drugs used for sedation and drugs studied, and this point should be discussed in the manuscript!

5.Lines 75-90: Was there any fasting or food restriction applied before or after drug administration? Need more information.

6.Lines 88-90: How were the tablets administered to dogs? Were they given with some water?

7.Line 102 and 108: Should be “Phoenix WinNonlin”.

8.Table 1: Please give bioavailability as “F (%)”

9.Line 168: “… these are no longer commercially available”. In the US?

10.There are large inter-individual variations for MAT and t1/2 values in the oral OFT group. The possible reasons should be discussed in the manuscript.

Reviewer #2: The manuscript is well written and planned. The study is well supported with the pharmacokinetic assessment and is recommended for publication.

1. What about the ethical issues regarding deploying canines for the present study?

2. The authors should specify the breed of each subject(canine) in the manuscript as only non-beagle is mentioned for some canines.

Reviewer #3: The manuscript is sound. However, the paper deals with outsourced facility-produced theophylline, a brief disclosure of the formulation is required. An account of previous PK studies should also be incorporated and compared.

6. PLOS authors have the option to publish the peer review history of their article (what does this mean?). If published, this will include your full peer review and any attached files.

Reviewer #1: **Yes: **Cengiz Gokbulut

Reviewer #2: No

Reviewer #3: No

---

## [Author Response · Author response to Decision Letter 0]

30 Nov 2021

We are grateful to the editor and reviewers for their suggestions and recommendations. We have done our best to address all concerns. Please see below for a point-by-point response.

Response to Editor’s Comments:

- Section headers have been reformatted

- File and Tables have been reformatted as well as in text citations of these

- In text references have been changed from parentheses to brackets

- Authors lists in the reference section have been reduced to a maximum of 6 authors per reference

- Figure file names have been updated and supplementary material has been divided into individual files (pdf for tables and tiff for figures)

- Title page updated

2. In your Methods section, please provide additional details regarding participant consent from the owners of the animals. In the ethics statement in the Methods and online submission information, please ensure that you have specified (1) whether consent was informed and (2) what type you obtained (for instance, written or verbal). If the need for consent was waived by the ethics committee, please include this information.

The following statement was added to the methods section “Written, informed consent was obtained from owners of all animals included” on lines 61-62.

Response to Reviewer Comments:

Reviewer #1: 

1. Lines 23-24: “… 503B outsourcing facility that manufactures veterinary drugs (OFT), …”, but in Lines 56-57: “… a 503B outsourcing facility-produced theophylline (OFT) in dogs”. Please revise the first abbreviation in the abstract.

This statement has been changed to “A new, 503B outsourcing facility-produced theophylline product (OFT) is available for veterinary use.” Lines 23-24.

2. Line 64: “At the time of the study…”. Before the study, the animals should not have taken any medication for a certain period!

The following statement has been added “and for at least two weeks prior” Line 65.

3. Lines 75-79: “This study was conducted in two phases …” or two-way crossover?

Changed to: “This study was a two-way crossover design with an intravenous phase and an oral phase, separated by a 7-day washout period.” Line 130.

4. Lines 79-82: There was a possibility of pharmacokinetic interaction between these drugs used for sedation and drugs studied, and this point should be discussed in the manuscript!

Thank you for this comment. This has been added to the limitations section of the discussion. See line 360-365.

5. Lines 75-90: Was there any fasting or food restriction applied before or after drug administration? Need more information.

No there was not. Medication was administered without food restriction as would typically be done in small animal practice. This information is included on lines 152-154. “Dogs had free access to water throughout the study and were fed their normal diet in twice daily rations. On the first morning of each phase, the diet was fed just after drug administration.”

6. Lines 88-90: How were the tablets administered to dogs? Were they given with some water?

The following was added “with a small amount of wet food” Lines 157-158.

7. Line 102 and 108: Should be “Phoenix WinNonlin”.

Corrected

8. Table 1: Please give bioavailability as “F (%)”

Changed

9. Line 168: “… these are no longer commercially available”. In the US?

Correct, this has been added.

10. There are large inter-individual variations for MAT and t1/2 values in the oral OFT group. The possible reasons should be discussed in the manuscript.

This has been added to the discussion. Lines 300-325.

Reviewer #2: 

1. What about the ethical issues regarding deploying canines for the present study?

Thank you for this point. We have added that we obtained written informed consent from owners of all dogs included in this study. See lines 98-99.

2. The authors should specify the breed of each subject(canine) in the manuscript as only non-beagle is mentioned for some canines.

This information is included in S1 Table.

Reviewer #3:

The manuscript is sound. However, the paper deals with outsourced facility-produced theophylline, a brief disclosure of the formulation is required.

Thank you for this point. The following statement has been added: “The OFT is formulated with an excipient having a polymeric backbone of cellulose in a specific ratio designed to regulate the release of theophylline from the tablet” Lin 125-127.

An account of previous PK studies should also be incorporated and compared.

Thank you for this suggestion. Stylistically, it is our preference to compare our results to previous PK studies within the context of discussing our results, as we have done throughout the first two paragraphs of the discussion. We have added additional comparisons in the added discussion of interindividual variability in the third paragraph. Lines 300-325.

---

## [Editor Report · Decision Letter 1]

22 Dec 2021

Pharmacokinetics of a 503B outsourcing facility-produced theophylline in dogs

PONE-D-21-32370R1

Dear Dr. Reinhart,

We’re pleased to inform you that your manuscript has been judged scientifically suitable for publication and will be formally accepted for publication once it meets all outstanding technical requirements.

Kind regards,

Kaisar Raza

Academic Editor

PLOS ONE
---

## [Editor Report · Acceptance letter]

28 Dec 2021

PONE-D-21-32370R1 

Pharmacokinetics of a 503B outsourcing facility-produced theophylline in dogs 

Dear Dr. Reinhart:

I'm pleased to inform you that your manuscript has been deemed suitable for publication in PLOS ONE. Congratulations! Your manuscript is now with our production department. 

Kind regards, 

on behalf of

Dr. Kaisar Raza 

Academic Editor

PLOS ONE